# MASS: Model-Agnostic Shift-Equivariant Downsampling

## Abstract

The performance of convolutional neural networks (CNNs) are thought to be insensitive to image shifts. However, recent studies have revealed that downsampling layers in CNNs result in inconsistent outputs for shifted input images. In this study, we present an approach for performing downsampling that ensures absolute shift equivariance. By employing model-agnostic downsampling method that leverages origin selection functions obtained from coordinate-independent statistics of the feature map, we can achieve perfect shift equivariance, while still adhering to the conventional downsampling procedures. Our method allows CNNs to exhibit both improved accuracy and perfect shift invariance for image classification, while also achieving shift equivariance in semantic segmentation benchmarks. Furthermore, we introduce a methodology for achieving shift equivariance without the need for any additional training process. This is accomplished by transferring pretrained weights and replacing existing layers with shift-equivariant counterparts. Additionally, we show that fine-tuning of the modified CNNs leads superior performance compared to previously proposed models.

## 1 Introduction

Shift equivariance in neural networks is frequently required in various situations. For example, in a typical visual recognition task, objects should be consistently identified regardless of where they appear in the image. A more specific but industrially important set of problems comprises identifying failure modes or defect types in otherwise massively periodic systems such as flash memories, other semiconductor chips such as CPUs and GPUs, display panels for TV and monitors, and CMOS image sensors. In such devices, the same or similar unit structure is repeated in two dimensions billions of times or more. Moreover, detecting and classifying fabrication defects consistently regardless of always-present misalignment between the device and the inspection camera is a daunting task. In this regard, shift invariance in deep neural networks is required.

Convolutional neural networks (CNNs) have shown excellent performance in fundamental computer vision tasks, especially in classification problems (He et al., 2016; Huang et al., 2017; Sandler et al., 2018; Liu et al., 2022). Most CNNs consist of convolutional layers, downsampling (or subsampling) layers, nonlinear activation function, and fully connected layers. Although CNNs have been regarded as inherently equivariant to shift operations (Goodfellow et al., 2016) and therefore invariant to it at the prediction level, recent research found that they are actually not shift-invariant owing to inconsistencies introduced by the downsampling layers (Zhang, 2019; Azulay & Weiss, 2019).

We propose **M**odel-**A**gnostic **S**hift-equivariant down(up)**S**ampling (**MASS**), a methodology aimed at making any down(up)sampling layer equivariant through image shifting, employing coordinate-independent array data statistics. This approach addresses the challenge of inconsistent results induced by image shifts. Our design inherently incorporates shift equivariance into the structure, ensuring that prediction results for both in-distribution and out-of-distribution images exhibit precise equivariance. The contributions of the present study are outlined as follows:

- We introduce novel methods to enforce *shift-equivariant downsampling* and *shift-equivariant upsampling* in a model-agnostic way. Any downsampling and upsampling layers become truly shift-equivariant, based on origin selection using input data statistics, independent of the absolute coordinates.

- We show that MASS is highly compatible with existing CNN architectures because it can be integrated into any downsampling layer in CNNs, including max-pooling, average pooling, and strided convolution. We further demonstrate that weight transfer from conventional models to shift-equivariant models can achieve absolute shift equivariance with marginal performance degradation.

- The performance of our method is validated for image classification (ImageNet and CIFAR-10), semantic segmentation (PASCAL VOC). Implementing MASS leads to 100% shift equivariance and shift invariance, as well as improved performance metrics with respect to other proposed downsampling techniques.

## 2 RELATED WORKS

In image recognition tasks, the desired property is to transform response equivariantly to the common image transformations. Transformation-equivariant and invariant neural networks are gaining in popularity. Recently, there have been many studies on equivariant CNNs. It should be noted that for the classification and object detection tasks, progress has been made for classification, segmentation, and detection of the neural network independent to scale (Xu et al., 2014; Sosnovik et al., 2019; Marcos et al., 2018), rotation (Dieleman et al., 2016; Cohen & Welling, 2016; Cheng et al., 2016; **?**), and group operations (Cohen & Welling, 2016; Romero et al., 2020; Xu et al., 2021).

CNNs are thought to be inherently shift-equivariant since convolution operations are shift-equivariant by nature (Goodfellow et al., 2016). However, several papers have pointed out that classic CNN models are sensitive to image shifts (Azulay & Weiss, 2019; Engstrom et al., 2019; Zhang, 2019; Manfredi & Wang, 2020). In fact, violation of the shift equivariance in convolution and downsampling operations was discussed in an earlier study (Simoncelli et al., 1992). More recently, Zhang (2019) discovered that a lack of shift equivariance occur because CNNs inherently ignore classical sampling theory. Therefore, the main reason for the absence of equivariance is due to the existence of a downsampling process in CNNs. In addition, even a single pixel shift can lead to a severe accuracy drop (Zhang, 2019; Azulay & Weiss, 2019; Engstrom et al., 2019).

To solve the violence of the shift equivariant, several approaches have been proposed. Data augmentation is a popular approach to learn equivariance during training stage (Engstrom et al., 2019; Benton et al., 2020). While careful augmentation strategies substantailly improves shift equivariance (Gunasekar, 2022), however, data augmentation only works on in-distribution data, and does not guarantee shift equivariance in out-of-distribution data.

Another line of research is to apply anti-aliasing low-pass filter, which originate from classical sampling theory. To avoid aliasing, it has been shown that applying Gaussian filter before the downsampling layers improves the accuracy, consistency, and stability with respect to shift operations (Zhang, 2019; Azulay & Weiss, 2019). Gaussian filters induce information spread to adjacent locations. This anti-aliasing concepts are extended to find content-aware filter weights by learning (Zou et al., 2020).

The first absolute shift-invariant method for image classification tasks are proposed by Chaman & Dokmanić (2020). They proposed a new type of pooling strategy, called adaptive polyphase sampling (APS). They use all downsampling layers with APS to achieve truly shift-invariance. While the selection of the polyphase components of APS is based on the $l_2$ norm, learnable polyphase sampling (LPS) generalize to select polyphase component via learnable parameters (Rojas-Gomez et al., 2022). Additionally, polyphase sampling concepts are extended to shift-equivariant upsampling, allowing perfect shift equivariance to semantic segmentation (Chaman & Dokmanić, 2021; Manfredi & Wang, 2020).

## 3 BACKGROUND

In this section, we briefly introduce several key concepts as background information for the subsequent discussion.

**Image shifts** In image shifting, various types of shift have been explored in prior research (Azulay & Weiss, 2019). A common practice involves applying a standard shift, which typically entails mov-

Table 1: **Downsampling layers in various CNN architectures for ImageNet-1K.** These architectures commonly have 5 downsampling layers, each with a stride of 2 (with the exception of ConvNeXt, which employs a stride of 4 for the initial strided convolutional layer).

| Layer | ResNet | DenseNet | MobileNet | ConvNeXt |
|-------|--------|----------|-----------|----------|
| 1 | StrideConv | StrideConv | StrideConv | StrideConv(s=4) |
| 2 | MaxPool | MaxPool | StrideConv | - |
| 3 | StrideConv | AvgPool | StrideConv | StrideConv |
| 4 | StrideConv | AvgPool | StrideConv | StrideConv |
| 5 | StrideConv | AvgPool | StrideConv | StrideConv |

ing the entire image horizontally and/or vertically, followed by cropping, are commonly employed. Another type of shift is known as zero-padding shift. Both standard and zero-padding shifts result in the loss of pixel-level information at the image boundary, a phenomenon known as the boundary effect. In this study, we only deal with the *circular shift*, where the pixel values at the edge of the image wrap around to the opposite corner. It is also useful to represent periodic systems (Chen et al., 2022). Our primary focus is on circular shifts because they allow us to isolate the impact of shift operations from the boundary effect. The information in the original input features is fully preserved at the pixel level. We can formally define the circular shift operation $S : \mathbb{R}^{H \times W \times C} \to \mathbb{R}^{H \times W \times C}$ as

$$S_{h,w}[n, m] = x[(n + h) \bmod H, (m + w) \bmod W] \tag{1}$$

**Downsampling** A downsampling is a function that reduces input dimensions while preserving task-related information (Boureau et al., 2010). Downsampling layers are popular components within neural networks to reduce computational complexity and memory requirement. Given input $x \in \mathbb{R}^{H \times W}$, a downsampling function $f : \mathbb{R}^{H \times W \times C} \to \mathbb{R}^{\frac{H}{s} \times \frac{W}{s} \times C'}$ reduces the spatial dimensions.

where $s \in N$ is the stride parameter, which is a crucial factor in the downsampling process. In modern CNNs, different architectural choices are made for the downsampling layers. Table 1 lists the downsampling layers utilized in popular architectures when applied to the ImageNet dataset, including ResNet (He et al., 2016), DenseNet (Huang et al., 2017), MobileNet (Sandler et al., 2018), and ConvNeXt (Liu et al., 2022). These architectures perform 32 times reduction in total image size through 5 downsampling operations (4 for ConvNeXt).

It should be noted that there are many downsampling layers other than the typical downsampling layer, e.g. $l_p$ pooling (Estrach et al., 2014), generalized learnable pooling (Lee et al., 2016), and softpool (Stergiou et al., 2021). Different downsampling operations work well in certain condition and architectures. Notice that all these methods commonly ignore classical sampling theory, resulting in variant result to the shifted images.

**Shift equivariance and invariance** Based on the circular shifts and downsampling functions, one can formally define the shift-equivariant function in the context of neural networks. A function $f$ is shift-equivariant is if

$$f(S(x)) = S'(f(x)) \tag{2}$$

where $S$ is the circular shift operation applied to the input x, and $S' : \mathbb{R}^{\frac{H}{s} \times \frac{W}{s} \times C'} \to \mathbb{R}^{\frac{H}{s} \times \frac{W}{s} \times C'}$ is the corresponding shift operation applied to the reduced feature map $f(x)$. Since downsampling function f changes the dimensions of feature map, the domain of $S$ and $S'$ is different. Shift invariance can be seen as a special case where $S'$ is an identity operation leading to $f(S(x)) = f(x)$. In modern CNNs, equivariant feature maps can be transformed into invariant features through global averaging pooling.

Shift-equivariant properties are maintained across layers in CNNs. The building blocks of CNNs are convolutional, pointwise activation functions, and batch normalization layers. If padding mode of convolutional layer is circular, all exhibit shift-equivariance. Therefore, the challenge in preserving shift equivariance lies in the downsampling layers.

**Polyphase sampling** The initial instance of perfect shift invariance in the image classification was reported in the work of Chaman & Dokmanić (2020). In this approach, the component is chosen using a predefined selection criterion. Formally, the polyphase sampling function is defined as

$$f_{PS}(x)\,[n, m] = x\left[sn + k_x^*, sn + k_y^*\right] \tag{3}$$

where $n, m \in \mathbb{Z}$ are indices and $k_x^*, k_y^*$ are polyphase components in x and y directions. The original version is called adaptive polyphase sampling (APS), which select $k^*$ using simple rule, which based on the largest $l_2$ norm of each polyphase component as selection criterion. The idea of APS was generalized to learnable polyphase sampling (LPS), determining specific polyphase in a learnable way (Rojas-Gomez et al., 2022). One can view polyphase sampling methods as examples of a pooling method. Therefore, we cannot exploit previously developed various downsampling layers, for example, max-pooling, average pooling, or strided convolution.

## 4 METHODS

Here, we introduce MASS, which is an *architecture-agnostic scheme for shift-equivariant downsampling*. The key idea is that to ensure that downsampling consistently occurs irrespective of the input image by applying a shift operation based on a selection function. The utilization of MASS enables CNNs to attain perfect shift equivariance while maintaining the integrity of the original downsampling operations.

### 4.1 SHIFT-EQUIVARIANT DOWNSAMPLING

**MASS scheme** We first set $(0, 0)$ as an origin of the unshifted input. In the case where the stride is $s$ in two-dimensional feature maps, there exist only $s^2$ unique sampling origins represented as $\mathbf{o} \in \{(0,0), (0,1), ..., (s,s)\}$. By consistently choosing the origin irrespective of input shifts, the downsampling outcomes remain unchanged. We define the *origin selection function* $h : \mathbb{R}^{H \times W \times C} \to \mathbb{R}^2$. Therefore, adhering to these instructions ensures the acquisition of pixels that are entirely identical.

1. select origin based on the predefined origin selection function. $\mathbf{o} = h(x)$
2. Shift image to fit the origin $\hat{x} = S_o(x)$
3. Conduct the original downsampling operation $y = f(\hat{x})$

Figure 1 illustrates the procedure for performing the downsampling process. The advantage of our approach is that the actual operation following origin selection remain exactly the same such as max-pooling, average pooling, and strided convolution. Therefore, we expect that the inherent effects of the existing downsampling will be preserved intact.

For simplicity, we use a simple hand-crafted selection function, which designates the polyphase component with the largest $l_2$ norm as the origin. It's worth noting that the selection rule can be a crucial factor in scenarios involving noise or boundary effects.

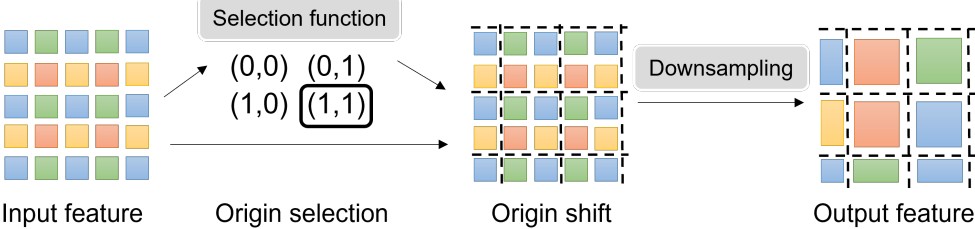

Figure 1: **Schematic illustration of MASS.** The selection function takes a feature map as input and produces an origin vector as output. This function consistently generates the same output regardless of shifts applied to the images.

**Trainable Parameters and Inference Time** incorporates two key steps alongside the downsampling operation: initially, the selection of an origin vector, and subsequently, the shifting of input features. Consequently, MASS operates as a deterministic process, devoid of any additional trainable parameters. The associated increase in inference time resulting from these additional steps is marginal.

## 4.2 COMPARISON WITH POLYPHASE SAMPLINGS

Polyphase sampling techniques can be regarded as particular pooling methods. When a polyphase component is chosen based on predefined criteria, it acquires data from precise locations within the pooling grid. On the contrary, MASS meticulously preserves the initial downsampling process. In the case of a feature map comprising a multi-channel array, the extraction of representative data can be achieved through a channel-wise approach.

Figure 2 provides visual representations of different downsampling methods. In the Max-pooling layer, the pooling process identifies the maximum components within a pooling window. When applied to shifted images, the result of the pooling varies because the origin of the pool remains unchanged. On the other hand, in the case of polyphase sampling, only data from specific regions is retained, which may result in potentially suboptimal representations. MASS-Max-pool combines the advantages of both selecting suitable representatives and ensuring shift equivariance.

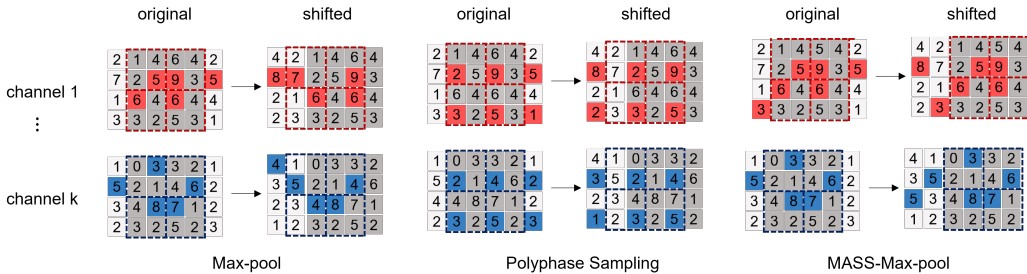

Figure 2: **Illustration of model-agnostic shift-equivariant downsampling.** Three separate downsampling operations are illustrated using multi-channel inputs.

## 4.3 WEIGHT TRANSFER FROM PRETRAINED NETWORK

The model-agnostic nature of MASS enables the direct utilization of pretrained weights. It allows for the seamless replacement of the original architecrues, facilitating the transformation of a CNN architecture into a perfectly shift-equivariant one. The lack of shift equivariance in conventional CNNs can be attributed to both the boundary effect and the downsampling operations. The schematic representation of this weight transfer process is depicted in Figure 3. Starting with the original CNN architecture, we introduce the following modifications: (1) we replace all padding within the convolution operation with circular padding, and (2) we substitute all downsampling layers with MASS-enabled layers. These adjustments do not impact the computations performed by the modified architectures.

## 4.4 SHIFT-EQUIVAIANT UPSAMPLING

Just as in the case of downsampling, we can extend MASS concept to upsampling operations. In the case of max-unpooling operations, MASS preserves both the indices and the origins of the corresponding max-pooling operations. It then utilizes origines for feature map shifts in reverse order to recover the original feature locations, and subsequently apply indices for unpooling. To facilitate comparison, Figure 4 illustrates three distinct approaches to upsampling. By incorporating MASS into both downsampling and upsampling processes, we can achieve genuine shift equivariance in CNNs that incorporate encoding-decoding architectures.architectures.

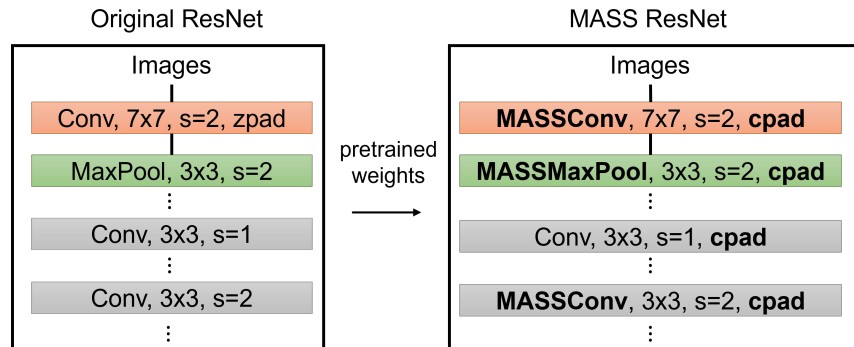

Figure 3: **Weight transfer from original ResNet to MASS ResNet architecture.** We replace all zero paddings (zpad) employed in convolutional layers with circular paddings (cpad), and we substitute all downsampling layers with MASS counterparts.

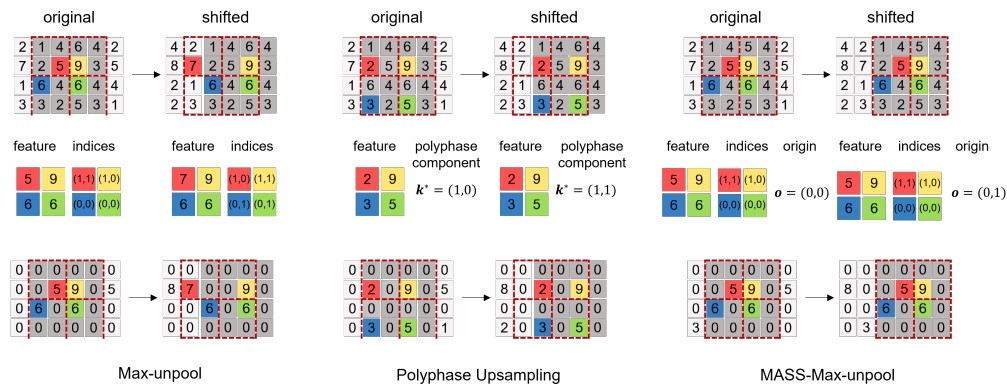

Figure 4: **Schematic of MASS for upsampling.** Demonstration of shift-equivariant upsampling method unpool as an example.

## 5 EXPERIMENTS

We perform benchmarks on typical vision tasks, specifically image classification and semantic segmentation. As discussed, we present results only for circular shifts to isolate the impact of shifts operations from boundary effect. Our method was implemented using Pytorch (Paszke et al., 2019).

### 5.1 IMAGE CLASSIFICATION

In image classification, we conducted experiments using small-scale datasets (CIFAR-10) and large-scale datasets (ImageNet-1K). We utilize Top-1 accuracy metric for both datasets, representing the percentage of correctly classified test/validation images by the model. To assess shift equivariance, we compute circular consistency, defined as:

$$\frac{1}{N} \sum_n \mathbf{1} \left[ y(S_{h_1, w_1}(x)) = y(S_{h_2, w_2}(x)) \right] \tag{4}$$

where $h_1$, $w_1$, $h_2$, and $w_2$ are randomly selected from the interval of {0,1,...,31}.

### 5.1.1 CIFAR-10

CIFAR-10 (Krizhevsky et al., 2009) is small image dataset comprising images with a resolution of $32 \times 32$ pixels. As our baseline models, we select two members chosen from ResNet: ResNet-20 and ResNet-18.

Table 2: **Image classification on CIFAR-10 test set.** Test accuracy and consistency on CIFAR-10 validation set. The reported values are obtained by averages and standard deviations from 5 independent runs.

| Method | Accuracy | | Consistency | |
|---|---|---|---|---|
| | ResNet-20 | ResNet-18 | ResNet-20 | ResNet-18 |
| Baseline | $90.47 \pm 0.30\%$ | $91.85 \pm 0.23\%$ | $90.94 \pm 0.45\%$ | $91.20 \pm 0.32\%$ |
| APS | $91.79 \pm 0.21\%$ | $94.58 \pm 0.07\%$ | **100%** | **100%** |
| MASS | $\mathbf{92.03 \pm 0.19}\%$ | $\mathbf{94.94 \pm 0.07}\%$ | **100%** | **100%** |
| LPF-3 | $92.05 \pm 0.18\%$ | $94.44 \pm 0.15\%$ | $91.20 \pm 0.32\%$ | $96.09 \pm 0.34\%$ |
| APS-LPF-3 | $92.20 \pm 0.21\%$ | $94.61 \pm 0.07\%$ | **100%** | **100%** |
| MASS-LPF-3 | $\mathbf{92.26 \pm 0.22}\%$ | $\mathbf{94.85 \pm 0.10}\%$ | **100%** | **100%** |
| DDAC-3 | $90.64 \pm 0.21\%$ | $94.40 \pm 0.24\%$ | $95.26 \pm 0.18\%$ | $97.21 \pm 0.07\%$ |
| APS-DDAC-3 | $92.36 \pm 0.14\%$ | $94.63 \pm 0.06\%$ | **100 %** | **100%** |
| MASS-DDAC-3 | $\mathbf{92.53 \pm 0.17}\%$ | $\mathbf{94.80 \pm 0.14}\%$ | **100 %** | **100%** |

Table 3: **Image classification on ImageNet-1K validation set.** We evaluate accuracy and consistency on the ImageNet validation set using pretrained weights. We compare the performance degradation of our method with the APS method, represented as delta.

| Model | Method | Top-1 Acc. | Delta | Consistency |
|---|---|---|---|---|
| | Baseline | 69.76% | - | 80.54% |
| ResNet-18 | APS | 51.23% | -18.53 | 99.99% |
| | MASS | 64.83% | **-4.93** | **100%** |
| | Baseline | 77.37% | - | 87.16% |
| ResNet-101 | APS | 64.90% | -12.47 | **100%** |
| | MASS | 72.71% | **-4.66** | **100%** |
| | Baseline | 74.5% | - | 85.9% |
| DenseNet-121 | APS | 34.2 % | -40.3 | **100%** |
| | MASS | 68.05% | **-6.45** | **100%** |
| | Baseline | 71.88% | - | 82.80% |
| MobileNetV2 | APS | 0.10% | -71.78 | **100%** |
| | MASS | 66.93% | **-4.95** | 100% |
| | Baseline | 84.12% | - | 92.87% |
| ConvNeXt-L | APS | - | - | - |
| | MASS | 83.39% | **-0.73** | **100%** |

**Training** To train the network, we used a learning rate of 0.2 for 250 epochs, with a batch size of 256, utilizing the stochastic gradient descent (SGD) method with momentum 0.9. We also used cosine annealing learning rate scheduling. A weight decay of 0.0005 per epoch was also applied. During the training stage, we augmented the original data by incorporating random horizontal flip augmentation.

**Anti-aliasing** The introduction of low-pass filters prior to downsampling can enhance shift equivariance because blurring helps prevent information loss during the downsampling process. As MASS is compatible with anti-aliasing filters, we conduct experiments with different filter applications. Two anti-aliasing filters are used: Gaussian filters, denoted as LPFs, and content-aware learned filters, called DDACs.

**Results** Table 2 represents the benchmark results on CIFAR-10 dataset. As results shows, MASS-enabled architectures consistently outperforms all other methods, including baseline and APS. Compared to baseline models without anti-alising filter, which shows 90.47% and 91.85%, our method improves 2.06% and 2.95%, respectively. Moreover, MASS also achieves circular consistency 100%.

The effectiveness of anti-aliasing filters has also been verified to improve accuracy. Among the models, MASS-DDAC-3, which applies MASS with a content-aware filter of size 3, demonstrates the best performance. These results indicate that our approach serves as a beneficial inductive bias in classification tasks.

Table 4: **Fine-tuning of MASS-enabled models.** After the weight transfer process, fine-tuning for 3 epochs enables the recovery of ResNet-18's accuracy. Results displayed above the middle line are sourced from Rojas-Gomez et al. (2022).

| Method | Top-1 Acc. | Consistency |
|---|---|---|
| Baseline | 64.88% | 80.39% |
| APS | 67.05% | **100%** |
| LPS | 67.39% | **100%** |
| DDAC-3 | 67.60% | **100%** |
| APS-DDAC-3 | 69.02% | **100%** |
| LPS-DDAC-3 | 69.11% | **100%** |
| MASS-Pretrain | 64.83% | **100%** |
| MASS-FT | **69.20%** | **100%** |

### 5.1.2 IMAGENET-1K

ImageNet (Deng et al., 2009) is a large image dataset comprising high-resolution images categorized into 1K classes for a classfication task. In our methodology, we do not train the networks from the beginning. Instead, we propose a weight transfer technique and demonstrate the efficacy of fine-tuning.

**Pretrained weight transfer**  Our primary discovery highlights the ability to harness the pretrained weights of CNNs through the use of MASS. Pretrained weights of CNN models are readily available, and for our experiments, we utilized PyTorch's official pretrained weights for various models. As demonstrated in Table 3, the straightforward replacement of the existing downsampling layer with our approach results in 100% consistency across all outcomes. In particular, in the case of the ConvNeXt-L model, we observe a mere 0.73% drop in accuracy.

We conducted a comparison between the weight transfer method of MASS and APS. In the official implementation of the APS model, all downsampling layers, including strided convolutions, are substituted with polyphase sampling layers. In all cases, the APS models significantly drop accuracy. Specifically, APS cannot replace downsampling layers within the ConvNeXt architecture because output channel number is different from input channel number in configuration of strided convolutions.

**Fine-tuning**  We propose a novel training strategy aimed at achieving higher accuracy with perfect shift equivariance in ImageNet classification. Inializing weight from pretrained weight, this strategy involves fine-tuning the modified model for 3 epochs with a learning rate set at 0.001. As demonstrated in Table 4 the fine-tuned model, MASS-FT, achieves a Top-1 accuracy of 69.22%, surpassing networks utilizing polyphase sampling even without any anti-aliasing filter. Compared to the original model, we observe only a marginal 0.56% decrease in accuracy, while exhibit 100% of circular consistency.

### 5.2 SEMANTIC SEGMENTATION

The majority of semantic segmentation architectures are generally consisting of an encoder followed by a decoder. To construct an encoder-decoder model that exhibit shift equivariance, it is essential to have MASS as complementary pairs.

We utilize two metrics for evaluating the performance of neural networks in semantic segmentation. The first metric is *mean intersection over union* (mIoU), a conventional measure used to gauge the accuracy of the segmentation task. To evaluate shift equivariance, we employ the second metric, *mean Average segmentation circular consistency* (mASCC), as introduced by Zou et al. (2020). mASCC is defined by

$$mASCC = \frac{1}{N} \sum_n \mathbf{1} \left[ S_{h_1,w_1}(x)_{i,j} = S_{h_2,w_2}(x)_{i,j} \right] \tag{5}$$

Table 5: **Semantic segmentation on PASCAL VOC validation set.** Results above the middle line are from Rojas-Gomez et al. (2022).

| Model | Method | mIoU | mASCC |
|---|---|---|---|
| DeepLabV3+ | Baseline | 70.03% | 95.42% |
| | APS | 72.37% | 96.70% |
| | LPS | 72.37% | **100%** |
| DeepLabV3+ | MASS | 72.30% | **100%** |

This metric computes the average percentage of predicted labels that demonstrate consistency when subjected to two random circular shifts. The shifts in the $x$ and $y$ directions, denoted as $h_1, w_1, h_2$, and $w_2$ are randomly selected from the interval $\{0,1,...,512\}$.

### 5.2.1 PASCAL VOC

In the context of semantic segmentation, we employ the PASCAL VOC 2012 dataset (Everingham et al., 2015). This dataset comprises 10,582 training images, including augmented samples, and 1,449 validation images. For our experiments, we utilize DeepLabV3+ models, which are characterized by an encoder-decoder architecture featuring various dilated convolutions. As the backbone, we employ the MASS version of ResNet-18. We use pretrain weight from official PyTorch implementation.

**Training** We adhere to the training recipe produced by the implementations by VainF (2020). The total number of iterations amounts to 30,000 with a batch size of 16. We utilize the SGD optimizer in conjunction with polynomial step size scheduling.

**Results** Our implementation successfully demonstrates perfect shift equivariance in the domain of semantic segmentation. In terms of the mIoU metric, the performance of our model without incorporate anti-aliasing filters is on par with polyphase sampling methods with anti-alising filters.

## 6 CONCLUSION

We propose MASS, a novel scheme that enables CNNs to attain perfect shift equivariance while preserving the existing downsampling and upsampling operations. This approach seamlessly integrates into existing CNN models, not only enforcing true shift equivariance but also enhancing generalization, resulting in improved accuracy compared to traditional architectures. By replacing the original operation with our MASS scheme, we can design popular CNN architectures with shift equivariance. With a minimal amount of fine-tuning, we can achieve a comparable level of accuracy. Looking ahead, our aim is to demonstrate that MASS can be extended to achieve perfect shift equivariance in a range of image recognition tasks, including object detection and instance segmentation.

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
