# OpenReview forum: "Model-Agnostic Shift-Equivariant Downsampling"
_ICLR.cc/2024/Conference — Submitted to ICLR 2024_

### Official Review · Reviewer_QNhG · 2023-10-31

**Soundness:** 4 excellent
**Presentation:** 3 good
**Contribution:** 3 good
**Rating:** 6
**Confidence:** 4

**Summary:**

The paper presents a strategy to achieve perfect equivariance in convolutional neural networks. That is, the model produces exactly the same output when the input image is shifted horizontally or vertically. This is achieved by preserving statistics of the positions of the downsampling process in the pooling layers of convolutional networks. The results indicate that the method works perfectly for downsampling (classification) and upsampling (segmentation) operations without the need to re-train the models.

**Strengths:**

* The problem is important and the paper is well motivated.
* The solution is simple and generic, can be applied to any convolutional model.
* The results are strong. 100% equivariance is achieved in all experiments, demonstrating the effectiveness of the proposed solution.
* The approach works on downsampling and upsampling paths of CNNs. The evaluation includes image classification and semantic segmentation.

**Weaknesses:**

Main comments:

* Equivariance is demonstrated only for inference (test) time. It is unclear how the method would facilitate equivariance during training. In other words, by implementing MASS in all pooling layers, what augmentations would be unnecessary when training a new model? The only experiments that involved training a model from scratch were conducted with the CIFAR dataset, but the augmentation procedure was not explained. More analysis of equivariance during training would be informative.
* In general, the explanation of the method has a few gaps that could be better presented and clarified. For instance, the paper indicates that previous work ignores classical sampling theory, but how MASS uses classical sampling theory is not explained later. Also, it is unclear what the authors mean by "MASS meticulously preserves the initial downsampling process". The introduction indicates that MASS uses input data statistics to select the origin, but these statistics are not clearly defined later. The procedure could be more formally presented to avoid confusions.
* The paper mentions that non-equivariant methods can display severe accuracy drops, but this does not seem to be reflected in the results. The consistency of other methods is usually above 80% and the classification rate remains high. If shifts are introduced randomly, the accuracy of a non-equivariant method can change every time. Reporting how the results change with the amount of shift introduced to break the classification of a model would be informative.

Other comments:
* It is unclear if the method results in computing or memory overhead (even if minimal, what additional operations / variables are added compared to regular pooling).
* Some acronyms are not clearly defined, such as APS, LPF and DDAC. Table 2 uses them extensively without citations or explanations to what they refer to exactly.
* Some minor typos: equivaiant, "architectures.architectures", experimants, inializing.

**Questions:**

* Does the proposed method remove the need for using certain augmentations during training?
* Are the accuracy results reported without shifts? The experimental procedure is unclear, please explain.
* Can you add results of how shifting affects accuracy in shift-sensitive models?

---

### Official Review · Reviewer_1mgo · 2023-11-01

**Soundness:** 2 fair
**Presentation:** 1 poor
**Contribution:** 2 fair
**Rating:** 3
**Confidence:** 4

**Summary:**

This paper addresses the problem of how to preserve shift-equivariance property for convolutional neural networks. Specifically, the authors merely consider circular shift operation over the input image sample, and simply extend an existing method APS (adaptive polyphase sampling, proposed by Anadi Chaman and Ivan Dokmanić in their CVPR 2021 work) by incorporating a pre-defined selection function for determining the origin which can accurately fit the shift operation. Experimental validation is conducted on image classification and semantic segmentation tasks.

**Strengths:**

+ The problem, i.e., how to preserve shift-equivariance property for convolutional neural networks, is critical.

+ The proposed method is simple and hand-crafted even though its implementation is not clear.

+ Comparative experiments are conducted on both image classification (with CIAFR-10 and ImageNet-1K datasets) and semantic segmentation (with PASCAL VOC dataset) tasks.

**Weaknesses:**

- The method and presentation.

In this work, although the authors addresses a fundamental research problem, how to preserve shift-equivariance property for convolutional neural networks, the proposed method called MASS is rather incremental, lacking new tech insights. To the best of my knowledge, MASS is merely a simple modification of existing work APS (adaptive polyphase sampling) proposed by Anadi Chaman and Ivan Dokmanić in their CVPR 2021 paper. Specifically, the authors use a pre-defined selection function for determining the origin with ASP which can accurately fit circular shift operations over the input image sample.  Generally, I have not seen any insightful differences against APS.

The presentation of the method is poor: 1) usually no explanations for notations and terms appeared in formulas; 2) no explanations/details on the formulation of the proposed MASS; 3) rather coarse descriptions for Figure 1 and Figure 2; 4) some sub-figures are wrong, e.g., two sub-figures for MASS in channel 1 of Figure 1 are not consistent to the others.

The writing of the paper is also poor. Please see my comments in "Others" part for details.

- The limitations.

The authors did not discuss on the limitations of the proposed method.

- The experiments.

Note that the authors claim that the pre-defined selection function for determining the origin can accurately fit the shift operation. However, the authors did not provide any details on how to implement it in experiments. This makes experimental comparison confusing.

Comparison is limited to APS.

Experiments are not convincing. On CIFAR-10 dataset, the proposed MASS brings very marginal gains to APS. On PASCAL VOC dataset, MASS performs worse than APS. However, on ImageNet-1K dataset, MASS is much better than APS. What are the root reasons?

There is no ablation to study how does the proposed method MASS work.

How about the performance of MASS under other shift operations to the input image sample instead of circular shift operations?

- Others.

The writing can be improved significantly. There exist numerous typos, grammar errors and inaccurate descriptions throughout the whole paper. Here, I just list some example errors in the "Related Works" section:

1. "Cheng et al.,2016; ?" -> an inaccurate citation;
2. "a lack of shift equivariance occur" -> "a lack of shift equivariance **occurs**";
3. "While careful augmentation strategies substantailly improves" -> "While careful augmentation strategies **substantially improve**";
4. "Another line of research is to apply anti-aliasing low-pass filter, which originate" -> "Another line of research is to apply anti-aliasing low-pass filter, which **originates**";
5. "This anti-aliasing concepts are" -> "This anti-aliasing **concept is**";
6. "The first absolute shift-invariant method for image classification tasks are proposed " -> "The first absolute shift-invariant method for image classification tasks **is** proposed";
7. "While the selection of the polyphase components of APS is based on the l2 norm, learnable polyphase sampling (LPS) generalize to select" -> "While the selection of the polyphase components of APS is based on the l2 norm, learnable polyphase sampling (LPS) **is generalized** to select";
8. Many citations are not formal, even to APS.

----------------------------------------------------- Post Rebuttal----------------------------------------------------------

As the authors did not provide any responses to my concerns, I downgrade my rating from "borderline reject" to "reject".

**Questions:**

Please refer to my detailed comments in "Weaknesses" for details.

---

> ### Comment · Reviewer_1mgo · 2023-12-02
> **Post rebuttal rating**
>
> As the authors did not provide any responses to my concerns, I downgrade my rating from "borderline reject" to "reject"

---

### Official Review · Reviewer_pcfU · 2023-11-06

**Soundness:** 3 good
**Presentation:** 3 good
**Contribution:** 3 good
**Rating:** 3
**Confidence:** 4

**Summary:**

The proposed work suggests a downsampling technique that is shift equivariant by equivariant origin alignment. This method is well suited for adapting to the existing layer and can utilize the weights of pre-trained models. Moreover, the proposed models outperformed other shift equivariant techniques without introducing more learnable parameters.

**Strengths:**

The proposed method achieves perfect shift equivariance and performs better in classification tasks, even without any learnable sampling parameters (compared to the LPS). It is also well-suited for reusing pre-trained weights.

**Weaknesses:**

1. The proposed method resembles building equivariant layers with canonical functions [a]. The proposed method can be seen as a special case of the mentioned work for shift equivarinace. This limits the contribution of the paper.

2. The benefit of the proposed technique compared to the existing method (APS, LPS) is poorly described.

3. The improvements are marginal.


a. Equivariance With Learned Canonicalization Functions

**Questions:**

1. Figure 2 caption: “On the other hand, in the case of polyphase sampling, only data from specific regions is retained, which may result in potentially suboptimal representations. MASS-Max-pool combines the advantages of selecting suitable representatives and ensuring shift equivariance.”— I do not entirely understand the statement. We can perform convolution with max filter followed by LPS. What is the extra benefit of MASS-Max-pool?
2. Section 4.1 “ there exist only $s^2$ unique sampling origins represented as o ∈ {(0, 0),(0, 1), ...,(s, s)}.” — should it be “{(0, 0),(0, 1), ...,(s-1, s-1)}.”?
3. While training from scratch as the MASS-Max-pool shifts the input to match the new calculated origin, does it likely introduce unwanted data augmentation? Especially if the pooling window is large.

---

### Official Review · Reviewer_1rzF · 2023-11-06

**Soundness:** 2 fair
**Presentation:** 2 fair
**Contribution:** 2 fair
**Rating:** 3
**Confidence:** 4

**Summary:**

The paper presents an approach to shift equivariant up- and downsampling in convolutional neural networks.

**Strengths:**

- The paper presents a very compelling motivation for shift equivariance based on industrial applications
- The paper seems to provide the shift-equivariant scheme that is consistent by design and also compatible with multiple architectures
- The proposed approach in principle could be used without fine-tuning if some accuracy loss is acceptable, unlike some other methods in the literature

**Weaknesses:**

- The description of theory is not at all clear, especially the part related to figures. Maybe if I read this paper 2 or 3 times, I will eventually understand what color schemes imply in Figures 1,2,4. But I want to be able to understand this from the first glance. Figures should serve the purpose of clarifying things and not making them more obscure. Without the explanation of what colors are supposed to signify and explain in the figures - it is impossible to quickly understand what they are supposed to clarify.
- The compelling motivation for shift equivariance is not supported by problem specific datasets. All experiments are done on generic datasets. CIFAR-10 does not seem to fit the motivation at all with its 32x32 images. It seems like a misfit for the purpose of the paper. I expect that the industrial applications involve high-resolution imagery. If authors can provide results on high-resolution datasets, especially from the industrial domain this will make the results a lot more compelling. There is a recent dataset described here: https://arxiv.org/pdf/2303.06673.pdf. I am sure that more search will reveal more datasets like this. I remember encountering similar problems on kaggle.
- Consistency metric defined in equation (4) does not make any sense to me. What does it measure, what is x and y? Is it pointwise pixel match, if so, why there is no summation over pixels? One of the closing brakets is missing.
- Baselines used in experimental tables are not explained well. As a result, the experiments do not seem persuasive
    - What is the reference for DDAC and LPF?
    - Why LPS is not included in Tables 2,3?
    - Why Table 3 does not contain same baselines as Table 2? It seems that a few of the baselines in Table 2 are very effective. It may well be that APS with enhancements presented in Table 3 might be as effective or better than MASS?
- Results in Table 2 are marginal and statistically insignificant. To me, the value of this result is approaching 0, because most confidence intervals overlap. It does not make sense to use bold font to signify the best model, when the best model is not significantly different than another model.
- From Tables 2-4, I do not see a decisive value of the proposed approach with respect to other approaches such as LPS-DDAC-3 or ASP-DDAC-3. Why do we need this approach, what is the value?
- I am not sure I see value in using the pretrained version of any of the approaches discussed in the experimental section. What is the point, can you explain in detail the actual use case? When the networks are fine-tuned properly, many of them achieve very similar results.
- Table 5 confirms previous concerns.

**Questions:**

- page 2: missing reference. "2016; ?), and group operations"
- page 6: typo "Schemetic of MASS" -> "Schematic of MASS"?
- is your approach compatible with visual transformers?

---

### Official Review · Reviewer_gzfx · 2023-11-11

**Soundness:** 4 excellent
**Presentation:** 2 fair
**Contribution:** 3 good
**Rating:** 8
**Confidence:** 4

**Summary:**

The authors propose and implement MASS, a method for downsampling that is exactly equivariant to shifts in images, making CNN networks exactly equivariant (or invariant) with respect to this symmetry in their downstream tasks (which are typically image classification). The method is implemented in a “model-agnostic way”, and can be used to invariantize pre-trained CNNs. Its performance is demonstrated on standard data sets.

**Strengths:**

The problem of imposing exact shift equivariance (in CNNs) is important in current literature and applications. The proposed solution is simple, robust, and easily applicable to pre-existing methods.

**Weaknesses:**

Main
* The algorithmic/mathematical presentation should be clearer. Occasionally notation appears that has not been precisely defined (e.g. $S_o$ on pg. 4 or the use of the $y$ variable). I specifically find Figure 1 hard to understand.

* A substantial effort is made to separate the proposed method from general polyphase sampling, but the exact reason behind the latter’s “performance degradation” should be explained more rigorously, as this is what would set it apart in applications.

Minor
* There are several typos, and all acronyms need to be defined upon first appearance.

**Questions:**

Regarding the choice of selection rule:
* In what way does the choice of function matter? Can it be determined adaptively, if the noise distribution is known?
* And finally, is it always possible to find a good function regardless of the level of noise?

---

### Meta-Review · Area_Chair_gTWr · 2023-12-05

**Metareview:**

The paper proposes a model-agnostic up-/down-sampling method that is exactly equivariant to translations (shifts) for images leveraging “origin alignment”.

The reviewers agree that while the motivation of the paper is clear, the presentation needs significant improvement, particularly the algorithmic description. Most importantly, reviewers 1rzF and 1mgo raised serious concerns about the choice of benchmarks, metrics, and baselines. Additionally, some of the presented gains are marginal and statistically negligible due to the stochastic nature of the training.  Finally, while reviewer gzfx was enthusiastic, they did not participate in the discussion nor provide further evidence to support their upbeat view of the paper vis-à-vis the other reviewers' less favorable opinions.

The authors did not provide any feedback, thus none of the concerns raised by the reviewers were addressed.

Despite the clear motivation, the paper remained unmodified during the rebuttal period, leaving the presentation lacking and many of the concerns regarding evaluation and benchmarking unresolved. I recommend to reject.

**Justification For Why Not Higher Score:**

The paper seems to have several evaluation issues, and it requires a rather large overhaul.
One reviewer provided a high-score, but they didn't engage on the discussion not provide further evidence substantiating their positive view in light of the other reviewers' less favorable views.

**Justification For Why Not Lower Score:**

N/A

---

### Decision · Program_Chairs · 2024-01-16

Reject